# Re-emergence and influencing factors of mountain-type zoonotic visceral leishmaniasis in the extension region of Loess Plateau, China

**Zhuowei Luo[1,2], Fenfen Wang[3], Zhaoyu Guo[1], Lulu Huang[1], Peijun Qian[1], Wenya Wang[1], Shenglin Chen[1], Yuanyuan Li[1], Peijun Zhang[3], Yi Zhang[1], Bin Wu[3], Zhengbin Zhou[1], Yuwan Hao[1]\*, Shizhu Li[1]\***

**1** National Key Laboratory of Intelligent Tracking and Forecasting for Infectious Diseases, National Institute of Parasitic Diseases at Chinese Center for Disease Control and Prevention,Chinese Center for Tropical Diseases Research; NHC Key Laboratory of Parasite and Vector Biology; WHO Collaborating Center for Tropical Diseases; National Center for International Research on Tropical Diseases, Shanghai, China, **2** Beijing Center for Disease Prevention and Control, Beijing Research Center for Preventive Medicine, Beijing, China, **3** Yangquan Center for Disease Control and Prevention, Yangquan, Shanxi, China

\* haoyw@nipd.chinacdc.cn (YH); lisz@chinacdc.cn, stoneli1130@126.com (SL)

**Data Availability Statement:** The authors confirm that all data underlying the findings are fully available without restriction. All relevant data are

## Abstract

### Objective

To understand the epidemiological distribution characteristics of mountain-type zoonotic visceral leishmaniasis (MT-ZVL) in Yangquan City, Shanxi Province, China, from 2006 to 2021, to explore the influencing factors leading to the re-emergence of the epidemic, and to provide a basis for the formulation of targeted control strategies.

### Methods

Case information spanning from 2006 to 2021 in Yangquan City was collected for a retrospective case-control study conducted from June to September 2022. A 1:3 matched ratio was employed. A questionnaire was utilized to gather data on basic information, demographic characteristics, awareness of MT-ZVL knowledge, residence, and dog breeding and living habits. The study employed a multifactorial conditional stepwise logistic regression model to analyze the influencing factors.

### Results

A total of 508 subjects was analyzed. Risk factors for MT-ZVL included the use of soil/stone/concrete as building materials ($OR = 3.932$), presence of nearby empty/stone stack houses ($OR = 2.515$), dog breeding ($OR = 4.215$), presence of stray dogs ($OR = 2.767$), and neighbor's dog breeding ($OR = 1.953$). Protective factors comprised knowledge of MT-ZVL ($OR = 0.113$) and using mosquito repellents ($OR = 0.388$). The findings indicate significant associations between environmental and behavioral factors and MT-ZVL incidence in Yangquan City, Shanxi Province, China, from 2006 to 2021. These results underscore the importance

within the paper and its Supporting Information files.

**Funding:** This work was supported by the National Key Research and Development Program of China (No. 2021YFC2300800, 2021YFC2300803); the National Natural Science Foundation of China (No. 32161143036, 32311540013); the International Joint Laboratory on Tropical Diseases Control in Greater Mekong Subregion (no. 21410750200) and the Shanxi Provincial Health Commission Project (2021021). The funders had no role in study design, data collection and analysis, decision to publish, or preparation of the manuscript.

**Competing interests:** The authors have declared that no competing interests exist.

of public awareness campaigns and targeted interventions aimed at reducing exposure to risk factors and promoting protective measures to mitigate the re-emergence of MT-ZVL outbreaks.

## Conclusion

House building materials, presence of neighboring empty houses, breeding domestic dogs and distribution of stray dogs surrounding the home are risk factors for MT-ZVL. Awareness of MT-ZVL and implementation of preventive measures during outdoor activities in summer and autumn are protective and may reduce the risk of MT-ZVL.

### Author summary

In recent years, a rapid reemergence of mountain-type zoonotic visceral leishmaniasis (MT-ZVL) has been seen in the extension region of the Loess Plateau that had previously achieved elimination in China. In this study, we investigate the epidemiological characteristics of MT-ZVL and identify the factors associated with the rapid rise in MT-ZVL epidemics using a case-control design. A questionnaire was used to collect basic information, demographic characteristics, awareness of MT-ZVL knowledge, residence, and dog breeding and living habits.

A multifactorial conditional stepwise logistic regression model was used to analyze the influencing factors. The study revealed that there has been a rising trend in the MT-ZVL endemic areas. House building materials, presence of neighboring empty houses, breeding domestic dogs and distribution of stray dogs surrounding the home are risk factors for MT-ZVL. Awareness of MT-ZVL and implementation of preventive measures during outdoor activities in summer and autumn are protective and may reduce the risk of MT-ZVL.

We believe that our study makes a significant contribution to the literature to develop strategy for control the ongoing local transmission of MT-ZVL in hill districts. By analyzing the epidemiological characteristics and influencing factors of MT-ZVL, this research provides a basis for further optimizing the targeted control strategies.

## Introduction

Leishmaniasis is a parasitic disease caused by *Leishmania* species and transmitted by sand flies. Three main forms of leishmaniasis that have been characterized include cutaneous leishmaniasis (CL), visceral leishmaniasis (VL), and mucocutaneous leishmaniasis (ML) [1–2]. VL, also known as Kala-azar, is a neglected tropical disease, which is the second most fatal parasitic disease after malaria. If not treated in a timely fashion, over 95% of VL patients die from complications within one to two years of onset. An estimated 50,000 to 90,000 new VL cases are diagnosed globally each year; only 25% to 45% are reported to the World Health Organization (WHO). More than 90% of new VL cases reported to WHO occurred in 10 countries in 2020 (Brazil, China, Ethiopia, Eritrea, India, Kenya, Somalia, South Sudan, Sudan, and Yemen). VL was targeted for elimination as a public health problem (EPHP) in 2020 [3–5].

In China, aproximately 530,000 VL patients were prevalent across 680 counties and cities out of 16 provinces, autonomous regions, and municipalities, north of the Yangtze River,

especially in northern Jiangsu and Anhui, Shandong, Henan, southern Hebei and Shaanxi plains, resulting in high mortality [6]. After the founding of the People's Republic of China, China had formulated the work plan to take the control and elimination efforts for a variety of serious diseases as the priority task of health work. This included nine diseases, among which VL were required to be eliminated within the time frame set by the "Framework of the National Agricultural Development Programme" in 1958 [7]. The number of patients decreased yearly, and the VL epidemic has been brought under control in the plain region of eight provinces/municipalities in the early 1960s. No new human infection has been found since 1983, reaching the goal of eliminating VL [8].

Epidemiologically, VL in China can be divided into three types, namely anthroponotic VL (AVL) in the plain region, mountain-type zoonotic VL (MT-ZVL) in the hilly region, and desert subtype of zoonotic VL (DST-ZVL) in the desert region [9]. MT-ZVL is a zoonotic infectious disease, and the source of human or livestock infection is dogs infected with *Leishmania*. Epidemics of VL have remained low in the 21st century. Nevertheless, the number of MT-ZVL cases reported in central China has rapidly increased and the geographic coverage of MT-ZVL epidemics has quickly expanded, with patients found predominantly in the Loess Plateau and its extensions, including Gansu, Shanxi, Shaanxi, Sichuan and Henan provinces [10–12]. Shanxi province has been historically endemic for MT-ZVL, with the most cases of MT-ZVL reported re-emergence of this subtype [13]. Several counties (districts) have become hyperendemic for MT-ZVL in China, with an apparent expanded distribution of MT-ZVL cases [14]. MT-ZVL cases are concentrated in Yangquan city, which is located in central and eastern Shanxi province and the eastern edge of the Loess Plateau. Mountainous and hilly terrains are predominant in the city [15]. Following concerted efforts in the 1950s, no local case had been reported since 1964 [16]. However, MT-ZVL re-emerged in Yangquan city in 2006, with sporadic cases occasionally reported. The number of reported MT-ZVL cases has continued to rise in Yangquan city since 2015, with a peak of 92 MT-ZVL cases in 2021[17–18].

This study aimed to investigate the epidemiological characteristics of MT-ZVL and identify the factors affecting the rapid rise in MT-ZVL epidemics using a case-control design in Yangquan city, Shanxi province. The intent was to inform the formulation of an MT-ZVL control strategy and improvements of the surveillance system.

## Materials and methods

### Ethics statement

Ethical review and approval were not required for the study on human participants in accordance with the local legislation and institutional requirements. Written informed consent from the patients/participants OR patients/participants legal guardian/next of kin was not required to participate in this study in accordance with the national legislation and the institutional requirements.

### Data source

VL patients reported in Yangquan city from January 1, 2006 to December 31, 2021 were retrieved from the Infectious Diseases Reporting Information Management System of China Information System for Disease Control and Prevention. Following exclusion of suspected, repetitive, and CL cases, all MT-ZVL patients with clinical and confirmed diagnoses with a current address in Yangquan city, Shanxi province were included. MT-ZVL was diagnosed according to *National Diagnostic Criteria for Visceral Leishmaniasis in China* [8]. Population data were collected from the *Shanxi Statistical Yearbook*. Vector county (district) and township

(street) maps in Yangquan city were provided by Yangquan Municipal Center for Disease Control and Prevention.

## Analysis of epidemiological features of MT-ZVL

Individual case data were captured from the surveillance of MT-ZVL in Yangquan city, Shanxi province, from 2006 to 2021 and were managed using Microsoft Excel 2007 (Microsoft Corporation; Redmond, WA, USA). The temporal, spatial, and population characteristics of MT-ZVL were descriptively analyzed.

## Identification of influencing factors for the rapid rise in MT-ZVL epidemics

A case-control study was performed. MT-ZVL cases were matched at a ratio of 1:3 with non-MT-ZVL patients with the same gender and age differences ≤3 years sampled from the same community (village) as controls. Presence of stray dogs surrounding homes was selected as a major exposure factor. Pretest results included an odds ratio ($OR$) = 2.15, $\alpha$ = 0.05, $\beta$ = 0.10, 45.6% rate of presence of stray dogs in the vicinity of the homes of controls, and 15% to 20% rates of lost responses to the questionnaire survey and unqualified questionnaires [19]. Based on these pretest results, the sample size estimated using PASS version 15.0 software was 372, including 93 subjects in the case group and 279 in the controls. Finally, a total of 508 subjects were included in this study, including 127 subjects in the case group and 381 in the control group, which met the needs of the sample size. The sample size ($n$) was estimated using the following formula:

$$n = \frac{\left[ Z_{1-\frac{\alpha}{2}}\sqrt{\left(1+\frac{1}{r}\right)\bar{P}(1-\bar{P})} + Z_{\beta}\sqrt{P_1\frac{(1-P_1)}{r} + P_0(1-P_0)} \right]^2}{(P_1 - P_0)^2}$$

A field questionnaire survey was performed during the period between June and September, 2022. The questionnaire was designed based on literature review and expert consultation, including demographics (age, gender, educational level, number of permanent residents, and occupation), incidence of MT-ZVL, and risk factors (basic information, demographic characteristics, awareness of MT-ZVL knowledge, residence, dog breeding, and living habits). Investigators were trained prior to questionnaire surveys and eligible investigators were recruited to participate in face-to-face questionnaire surveys.

## Statistical analyses

All epidemiological data of MT-ZVL were entered into Microsoft Excel 2016 (Microsoft Corporation). A Joinpoint regression model was created using Joinpoint Regression Program version 4.3.1 software. The trends in incidence of MT-ZVL were examined using annual percent change (APC) in Yangquan city from 2006 to 2021. Univariate and multivariate logistic regression analyses were performed to identify factors affecting the rapid rise in MT-ZVL epidemics. All data were double entered into EpiData version 3.1 software, and all statistical analyses were performed using SPSS version 22.0 and R package version 4.0 software. Continuous variables with normal or approximate normal distribution were expressed as mean ± standard deviation (SD), and categorical variables were described as frequency or proportion. Factors affecting the incidence of MT-ZVL were screened with univariate and multivariate logistic regression models, and each independent variable was subjected to univariate logistic regression analysis. Binary variables were assigned 0 or 1, while multi-categorical variables were assigned with

dummy variables. Based on results of univariate analysis (independent variables with $P < 0.01$) and professional knowledge, all potential influencing factors were included in the multivariate conditional stepwise logistic regression model, the strength of correlation was examined using *OR* and its 95% confidence interval (*CI*). A two-tailed $P < 0.05$ was considered statistically significant.

## Results

### Epidemiological features of MT-ZVL

Few MT-ZVL cases were reported in Yangquan city, Shanxi province from 2006 to 2014, including one case each in 2006, 2008, 2010, 2011, 2013, and 2014. The number of reported MT-ZVL cases appeared to rise with time since 2015. A total of 254 cases was reported from 2015 to 2021, with an annual mean incidence of 2.75 cases per 100,000 population. Joinpoint regression analysis showed a tendency towards a rapid rise in incidence of MT-ZVL in Yangquan city from 2015 to 2021 [APC = 56.3%, 95% *CI*: (34.4%, 81.8%), $t = 7.6$, $P < 0.01$] (Fig 1).

From 2006 to 2021, MT-ZVL occurred at all age groups in Yangquan city, predominantly among elderly residents aged ≥50 years (47.02%, 122/260). Among patients ≥50 years, MT-ZVL cases were predominant among those ≥60 years of age (31.92%, 83/260). The patients included 181 men and 79 women, with a male-to-female ratio of 2.29:1. Farmers were the most common occupation (41.15%, 107/260), followed by diaspora children (21.92%, 57/260) and housework/unemployed (11.92%, 31/260).

Local MT-ZVL cases were reported across all counties (districts) of Yangquan city. From 2015 to 2019, the largest number of MT-ZVL cases was reported in Jiaoqu district, with an obvious rise in the number of reported MT-ZVL cases in 2020 and 2021 compared to 2019, and a higher number of local cases in Pingding county than in Jiaoqu district. The high-incidence regions included Pingding county and Jiaoqu district in Yangquan city until 2021. A

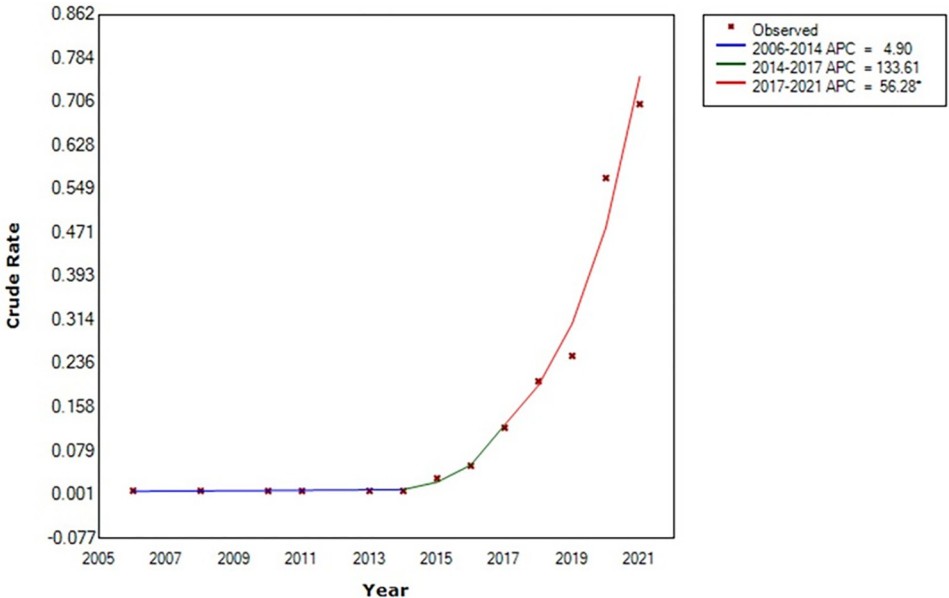

\* Indicates that the Annual Percent Change (APC) is significantly different from zero at the alpha = 0.05 level.
Final Selected Model: 1 Joinpoint.

**Fig 1. Incidence trend of MT-ZVL in Yangquan city from 2006 to 2021using the Joinpoint regression model.**

relatively low incidence rate of MT-ZVL was found in Yuxian county. However, re-emergence of MT-ZVL occurred in recent years.

## Factors affecting the incidence of MT-ZVL

A total of 508 subjects (71.6% men) were enrolled. They included 127 cases in the case group and 381 in the control group who were matched by gender, age, place of residence. Subjects in the case group were recruited from 84 villages/communities/streets in 23 townships from Chengqu district, Jiaoqu district, Kuangqu district, Pingding county, and Yuxian county (Fig 2). The participants in the case group had a mean age of 48.91 ± 22.919 years. Of the 127 individuals, men (71.7%, n = 91), individuals living with three to four family members (46.5%, n = 59), junior high school educational level (38.6%, n = 49), and farmers (26.0%, n = 33) were predominant. There were no significant differences between the case and control groups in terms of gender, age, number of persons living in the family, educational level, or occupation ($P > 0.05$; Table 1).

Independent variables with $P < 0.01$ in the univariate analysis were included in the multivariate conditional stepwise logistic regression model. Factors associated with a higher risk of developing MT-ZVL included use of soil/stone/concrete as building materials [$OR = 3.932$, 95% $CI$: (1.278, 12.102), $P = 0.017$] presence of neighboring empty or stone stack houses compared to lack of empty neighboring houses $OR = 2.515$, 95% $CI$: (1.346, 4.699), $P < 0.01$],

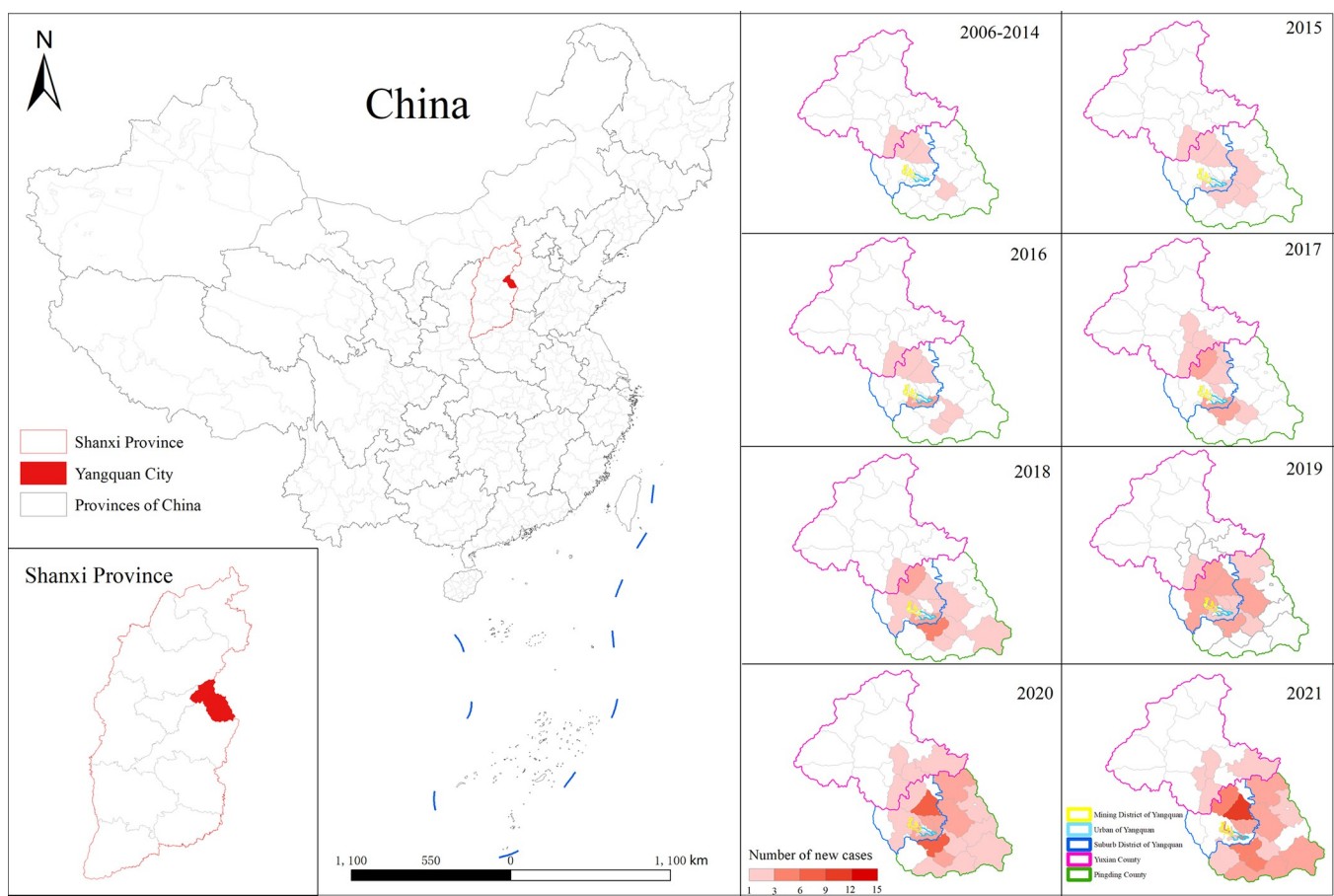

**Fig 2. Regional distribution of MT-ZVL cases in Yangquan City from 2006 to 2021 (Note: The map is based on the standard map downloaded from PlaniGlobe website (http://www.planiglobe.com), and the base map has not been modified).**

**Table 1. Demographic characteristics of MT-ZVL case-control subjects in Yangquan city.**

| Variable | Total | Case | Control | P value |
|---|---|---|---|---|
| **Gender** | | | | 1.000 |
| Male | 364(71.6%) | 91(71.7%) | 273(71.7%) | |
| Female | 144(28.4%) | 36(28.3%) | 108(28.3%) | |
| **Age** | | | | 0.514 |
| <5 | 60(11.8%) | 16(12.6%) | 44(11.5%) | |
| 5–14 | 24(4.7%) | 4(3.1%) | 20(5.2%) | |
| 15–39 | 26(5.1%) | 9(7.1%) | 17(4.5%) | |
| ≥40 | 398(78.4%) | 98(77.2%) | 300(78.7%) | |
| **No. of inhabitants** | | | | 0.417 |
| 1–2 | 204 | 51(40.2%) | 153(40.2%) | |
| 3–4 | 218 | 59(46.5%) | 159(41.7%) | |
| ≥5 | 86 | 17(13.4%) | 69(18.1%) | |
| **Educational level** | | | | 0.405 |
| No education | 102 | 28(22%) | 74(19.4%) | |
| Primary school | 115 | 30(23.6%) | 85(22.3%) | |
| Junior high school | 183 | 49(38.6%) | 134(35.2) | |
| Senior high school/technical secondary school/technical school | 76 | 16(12.6%) | 60(15.7%) | |
| Junior college and above | 32 | 4(3.1%) | 28(7.3%) | |
| **Occupation** | | | | 0.868 |
| Herdsman/farmer | 119 | 33(26.0%) | 86(22.6%) | |
| Employee | 83 | 21(16.5%) | 62(16.3%) | |
| Cadre | 11 | 2(1.6%) | 9(2.4%) | |
| Businessman | 6 | 0(0.0%) | 6(1.6%) | |
| Student | 13 | 3(2.4%) | 10(2.6%) | |
| Preschool children | 26 | 4(3.1%) | 22(5.8%) | |
| Diaspora children | 51 | 15(11.8%) | 36(9.4%) | |
| Housework/unemployed | 121 | 30(23.6%) | 91(23.9%) | |
| Others | 78 | 19(15.0%) | 59(15.5%) | |

Awareness of MT-ZVL, housing conditions, dog breeding, and personal living habits were included in the univariate analysis. The analysis revealed significant differences between the case and control groups in terms of knowledge of MT-ZVL, how this knowledge was acquired, house building conditions, installation of fine-pore window screens, presence of neighboring unoccupied stone fold or empty house, breeding dogs, presence of stray dogs in the vicinity, presence of breeding dogs in neighboring houses, mosquito deterrents including the use of mosquito repellent incense, wearing long-sleeved jackets, or use of mosquito repellent at evening during summer or autumn, or at night outdoor activities ($P < 0.05$; Table 2).

breeding dogs compared to no dog breeding [$OR$ = 4.215, 95% $CI$: (2.027, 8.763), $P < 0.01$], presence of stray dogs in the vicinity compared to the absence of stray dogs [$OR$ = 2.767, 95% $CI$: (1.5, 5.103), $P < 0.01$], and breeding dogs in a neighboring house compared to no dog breeding [$OR$ = 1.953, 95% $CI$: (1.05, 3.633), $P$ = 0.035]. Protective factors for MT-ZVL included knowledge of MT-ZVL [$OR$ = 0.113, 95% $CI$: (0.059, 0.219), $P < 0.01$] and use of mosquito-repellent incense, wearing long-sleeved jackets or use of mosquito repellent at evening during summer or autumn, or at night outdoor activities [$OR$ = 0.388, 95% $CI$: (0.187, 0.805), $P$ = 0.011] (Fig 3).

## Discussion

MT-ZVL has been historically prevalent in Shanxi province. This parasitic disease was widely prevalent in Shanxi province in the early 1950s. Disease transmission was almost under control

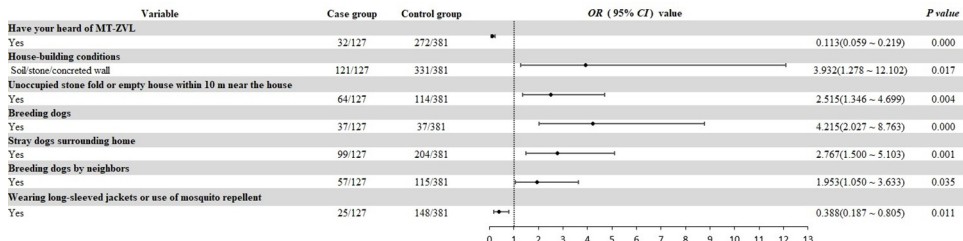

| Variable | Case group | Control group | OR (95% CI) value | OR (95% CI) value | P value |
|---|---|---|---|---|---|
| **Have your heard of MT-ZVL** | | | | | |
| Yes | 32/127 | 272/381 | | 0.113(0.059 ~ 0.219) | 0.000 |
| **House-building conditions** | | | | | |
| Soil/stone/concreted wall | 121/127 | 331/381 | | 3.932(1.278 ~ 12.102) | 0.017 |
| **Unoccupied stone fold or empty house within 10 m near the house** | | | | | |
| Yes | 64/127 | 114/381 | | 2.515(1.346 ~ 4.699) | 0.004 |
| **Breeding dogs** | | | | | |
| Yes | 37/127 | 37/381 | | 4.215(2.027 ~ 8.763) | 0.000 |
| **Stray dogs surrounding home** | | | | | |
| Yes | 99/127 | 204/381 | | 2.767(1.500 ~ 5.103) | 0.001 |
| **Breeding dogs by neighbors** | | | | | |
| Yes | 57/127 | 115/381 | | 1.953(1.050 ~ 3.633) | 0.035 |
| **Wearing long-sleeved jackets or use of mosquito repellent** | | | | | |
| Yes | 25/127 | 148/381 | | 0.388(0.187 ~ 0.805) | 0.011 |

**Fig 3. Multivariate conditional logistic regression analysis of influencing factors of MT-ZVL.**

at the end of 1959 following massive control programs [20–21]. Few cases have been reported each year since 1960 [21]. The study site of Yangquan city (11254′ to 11404′ E, 3740′ to 3831′ N) is located in eastern Shanxi province, west of the middle section of Taihang Mountain and eastern edge of the Loess Plateau. Mountainous terrains are predominant in the city, which measures 3362.1 hm², and there are also hills and plains. The city has a warm, temperate, semi-humid, continental monsoon climate.

Gradual improvements in ecological environments and continuous socioeconomic developments preluded a rapid rise in the number of locally reported MT-ZVL cases in Yangquan city since 2015, including a local outbreak. In addition, local MT-ZVL cases occurred across all five counties (districts) of Yangquan city in 2020, with a total of 75 cases reported (35% of the total reported cases in China). The incidence of MT-ZVL in Yangquan city ranked first in China. The annual mean incidence of MT-ZVL was 2.75 cases per 100,000 population from 2015 to 2021. The reported MT-ZVL cases had a male-to-female ratio of 2.29:1 in Yangquan city from 2006 to 2021, which was similar to that in Shanxi province from 2010 to 2019 [22]. The cases occurred predominantly among the elderly ≥60 years of age and diaspora children <3 years of age. Previous studies have demonstrated that MT-ZVL occurs predominantly among children <10 years of age, and is highly prevalent among diaspora children and farmers, which is inconsistent with the findings from this study [23]. This discrepancy may be attributed to socioeconomic development in Yangquan city. The population structure has changed in rural areas, with elderly resident becoming predominant as children and their parents have migrated into urban areas. Most of the elderly residents suffer from underlying diseases, and have become highly prevalent populations for MT-ZVL. Our data revealed that most MT-ZVL cases were concentrated in hilly regions at the border between urban and suburb areas of Yangquan city, with approximately 70% of total cases reported in Jiaoqu district and Pingding county. However, MT-ZVL cases appeared with a dot-like distribution, with no remarkable clusters. In addition, the MT-ZVL epidemics displayed a remarkable tendency towards expansion [24–25]. MT-ZVL cases have been reported in Yuxian county, north of Yangquan city, since 2019, and the number of townships affected by MT-ZVL has tended to rise annually [18].

The development of MT-ZVL has been associated with building materials, dog breeding, and personal living habits [26–28]. However, the factors affecting the re-emergence of MT-ZVL in Yangquan city have been unclear. In this study, most of the participants were recent MT-ZVL patients, who had accurate memories of potential risk factors they were exposed to. We performed a 1:3 matched case-control study to eliminate the potential effects of region, gender, and age. Our findings show that house building materials, presence of empty neighboring houses, breeding domestic dogs, presence of stray dogs in the vicinity of the subject's home, knowledge of MT-ZVL, and implementation of preventive measures during outdoor activities in summer or autumn were associated with the development of MT-ZVL.

To identify the factors affecting MT-ZVL incidence in Yangquan city, a total of 508 MT-ZVL cases were enrolled in the case-control study, including 127 subjects in the case group and 381 in the control group. No significant differences were evident between these groups in terms of gender, age, or place of residence. Univariate analysis revealed that knowledge of MT-ZVL was 59.8% among participants, with higher awareness in the control group than in the case group. The majority (64.6%) of the participants in the case group had no route for acquiring MT-ZVL-associated information. Short videos, as a novel publicity tool, had been satisfactorily accepted and popularized (14.2%). Our findings showed that both building materials of the house and breeding dogs correlated with the development of MT-ZVL; the type of night outdoor activities in summer or autumn and eating in courtyards were not included in the multivariate logistic regression model. Use of soil/stone/concrete as building materials was associated with a higher risk of developing MT-ZVL than use of ceramic tiles/whitewashing, consistent with previous reports [23]. Soil/stone/concrete are common building materials in local cave dwellings. The surfaces of these materials are not flat, full of cracks, and are warm in winter and cool in summer, which provides a favorable condition for sand flies to hide, which increases the likelihood of human exposure to sand fly bites [29]. The risk of MT-ZVL was 3.9 times higher among residents of houses built using soil/stone/concrete as building materials than in houses constructed of ceramic tiles/whitewashing. The presence of empty houses surrounding the subject's house conferred a 2.5 times higher risk of MT-ZVL than the absence of empty houses. Most empty houses have no residents and are full of wild grasses, which can provide breeding sites for local sand flies [30]. The flies can enter neighboring houses at dusk seeking a blood meal, which greatly increases the risk of MT-ZVL. Breeding dogs is a widely accepted risk factor for MT-ZVL, and the present results validate this belief. Dogs are an important reservoir host and source of infection in regions endemic for MT-ZVL [31–32]. Domestic dogs are in close contact with humans, and the nested PCR assay results have revealed a 32% to 64% prevalence of canine *Leishmania* infection in areas endemic for MT-ZVL [33]. These data indicate that breeding dogs is a risk factor for human MT-ZVL. In the present study, the presence of stray dogs in the vicinity of a subject's house increased the risk of human MT-ZVL, suggesting that the widespread activities of stray dogs may greatly increase the coverage of sand fly bites, resulting in transmission and spread of MT-ZVL. This view is consistent with previous reports showing a higher risk of *Leishmania* infection in dogs from the edge of villages than from the center of villages [23]. In addition, our findings showed that knowledge of MT-ZVL, wearing long-sleeved jackets and trousers, and use of mosquito repellent during outdoor activities in summer and winter were protective factors for MT-ZVL. The awareness of MT-ZVL knowledge was 59.8% in the control group, while only 25.2% subjects in the case group had heard of MT-ZVL, which was similar to the questionnaire survey results in Dangchang county, Gansu province (27.9%). Only 19.7% of subjects in the case group wore long-sleeved jackets and used mosquito repellent at dusk or during outdoor night activities in summer and autumn, indicating poor understanding of MT-ZVL and low awareness of MT-ZVL preventive measures among residents living in endemic foci. Therefore, improved health education is important for MT-ZVL control in endemic areas.

This study has some limitations. First, the data presented in this study were from a case-control study, and the time frame of cases was very wide which fails to identify the causal relationship. Because of limitations of a case-control design, recall bias and reporting bias cannot be excluded. Second, while most confounding factors were considered in this study, multiple factors affect the development of MT-ZVL. All confounding factors cannot be controlled during data analysis, and further studies with directed acyclic graph are needed to examine the causal relationship between variables with topological graphs and manage potential bias.

**Table 2. Univariate analysis of a case-control study of MT-ZVL in Yangquan city.**

| Variable | Total (*n* = 508) | Case group (*n* = 127) | Control group (*n* = 381) | *P* value |
|---|---|---|---|---|
| **Awareness of MT-ZVL** | | | | |
| Have your heard of MT-ZVL | | | | |
| No | 204 (40.2%) | 95(74.8%) | 109(28.6%) | |
| Yes | 304 (59.8%) | 32(25.2%) | 272(71.4%) | 0.000 |
| Route for acquiring MT-ZVL related knowledge | | | | |
| health professionals | 121 (23.8%) | 19(15.0%) | 102(26.8%) | 0.000 |
| Friends and neighbors | 74 (14.6%) | 12(9.4%) | 62(16.3%) | 0.000 |
| Broadcast | 34 (6.7%) | 4(3.1%) | 30(7.9%) | 0.000 |
| Brochures | 37 (7.3%) | 4(3.1%) | 33(8.7%) | 0.006 |
| Electronic media | 72 (14.2%) | 6(4.7%) | 66(17.3%) | 0.003 |
| No | 170 (33.5%) | 82(64.6%) | 88(23.1%) | 0.000 |
| **Housing conditions** | | | | |
| House-building conditions | | | | |
| Ceramic tiles/whitewashing | 56 (11.0%) | 6(4.7%) | 50(13.1%) | |
| Soil/stone/concreted wall | 452 (89.0%) | 121(95.3%) | 331(86.9%) | 0.029 |
| Installation of fine-pore window screens | | | | |
| No | 30(5.9%) | 14(11.0%) | 16(4.2%) | |
| Yes | 478(94.1%) | 113(89.0%) | 365(95.8%) | 0.016 |
| Unoccupied stone fold or empty house within 10 m near the house | | | | |
| No | 330(65.0%) | 63(49.6%) | 267(70.1%) | |
| Yes | 178(35.0%) | 64(50.4%) | 114(29.9%) | 0.000 |
| **Dog breeding** | | | | |
| Breeding dogs | | | | |
| No | 434(85.4%) | 90(70.9%) | 344(90.3%) | |
| Yes | 74(14.6%) | 37(29.1%) | 37(9.7%) | 0.000 |
| Stray dogs surrounding home | | | | |
| No | 205(40.3%) | 28(22.1%) | 177(46.5%) | |
| Yes | 303(59.7%) | 99(77.9%) | 204(53.5%) | 0.000 |
| Breeding dogs by neighbors | | | | |
| No | 336(66.1%) | 70(55.1%) | 266(69.8%) | |
| Yes | 172(33.9%) | 57(44.9%) | 115 (30.2%) | 0.009 |
| **Living habits** | | | | |
| Use of mosquito repellent incense/nets for prevention of mosquito bites | | | | |
| No | 259 (51.0%) | 79(62.2%) | 180(47.2%) | |
| Yes | 249 (49.0%) | 48(37.8%) | 201(52.8%) | 0.000 |
| Type of night activities in summer and autumn | | | | |
| Indoors | 215(42.3%) | 44(34.6%) | 171(42.3%) | 0.215 |
| Outdoor resting or walking | 286(56.3%) | 80(63.0%) | 206(56.3%) | 0.468 |
| Others | 7(1.4%) | 3(2.4%) | 4(1.4%) | |
| Eating in the courtyard at night in summer and autumn | | | | |
| No | 346(68.1%) | 80(63.0%) | 266(69.9%) | |

(*Continued*)

**Table 2.** (Continued)

| Variable | Total (*n* = 508) | Case group (*n* = 127) | Control group (*n* = 381) | *P* value |
|---|---|---|---|---|
| Yes | 162(31.9%) | 47(37.0%) | 115(30.1%) | 0.217 |
| Wearing long-sleeved jackets or use of mosquito repellent at evening during summer or autumn, or at night outdoor activities | | | | |
| No | 335(66.0%) | 102(80.3%) | 233(61.2%) | |
| Yes | 173(34.0%) | 25(19.7%) | 148(38.8%) | 0.001 |

In summary, the results of the present study demonstrate that the building materials of houses, presence of empty houses surrounding the subject's home, breeding domestic dogs, and distribution of stray dogs surrounding the home are risk factors for MT-ZVL. Awareness of MT-ZVL knowledge and implementation of preventive measures during outdoor activities in summer and autumn are protective factors for MT-ZVL, which may reduce the risk of MT-ZVL. Pesticide residue spraying in houses and empty houses surrounding the home, reduction of sand fly population density and widespread use of pesticide-immersed dog collars are required to interrupt the transmission of MT-ZVL between dogs and sand flies. In addition, dog management, such as reduction of the number of domestic and stray dogs, is needed in endemic foci of MT-ZVL, for prevention of disease expansion, and public health education is required to increase the self-protective awareness and timely healthcare-seeking awareness.

## Supporting information

**S1 Data. Excel spreadsheet containing numerical incidence and cases of MT-ZVL from 2006 to 2021 in data and statistical analysis for Fig 1 of the study.**
(CSV)

**S2 Data. Regional distribution of MT-ZVL cases in Yangquan City from 2006 to 2021 in data and statistical analysis for Fig 2 of the study.**
(CSV)

**S3 Data. The investigation data examined to determine risk factors of MT-ZVL in Yangquan City in statistical analysis for Table 1 and Table 2, and Fig 3 of the study.**
(XLSX)

## Acknowledgments

We thank everyone who cooperated with our investigation. We thank the staff of National Institute of Parasitic Diseases, Chinese Center for Disease Control and Prevention (Chinese Center for Tropical Diseases Research).

## Author Contributions

**Conceptualization:** Zhengbin Zhou, Yuwan Hao, Shizhu Li.

**Formal analysis:** Zhuowei Luo, Zhaoyu Guo, Lulu Huang.

**Funding acquisition:** Fenfen Wang, Shizhu Li.

**Investigation:** Zhuowei Luo, Fenfen Wang, Peijun Zhang.

**Methodology:** Zhuowei Luo, Yuanyuan Li, Zhengbin Zhou.

**Project administration:** Peijun Qian, Wenya Wang, Shenglin Chen.

**Supervision:** Yi Zhang, Bin Wu, Zhengbin Zhou.

**Writing – original draft:** Zhuowei Luo, Zhengbin Zhou.

**Writing – review & editing:** Yi Zhang, Yuwan Hao, Shizhu Li.

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
