## [Decision Letter · Decision Letter 0]

31 Jan 2024

Dear Dr. Li,

Thank you very much for submitting your manuscript "Re-emergence and influencing factors of mountain-type zoonotic visceral leishmaniasis in the extension region of Loess Plateau, China" for consideration at PLOS Neglected Tropical Diseases. As with all papers reviewed by the journal, your manuscript was reviewed by members of the editorial board and by several independent reviewers. In light of the reviews (below this email), we would like to invite the resubmission of a significantly-revised version that takes into account the reviewers' comments. 

We cannot make any decision about publication until we have seen the revised manuscript and your response to the reviewers' comments. Your revised manuscript is also likely to be sent to reviewers for further evaluation.

Sincerely,

Johan Van Weyenbergh

Academic Editor

Dileepa Ediriweera

Section Editor

Reviewer's Responses to Questions

**Key Review Criteria Required for Acceptance?**

**Methods**

-Are the objectives of the study clearly articulated with a clear testable hypothesis stated?

-Is the study design appropriate to address the stated objectives?

-Is the population clearly described and appropriate for the hypothesis being tested?

-Is the sample size sufficient to ensure adequate power to address the hypothesis being tested?

-Were correct statistical analysis used to support conclusions?

-Are there concerns about ethical or regulatory requirements being met?

Reviewer #1: The objectives of the study clearly and study design appropriate , the sample size sufficient.

Reviewer #2: Yes

Reviewer #3: Thank you for providing detail description of the methodology. I have a few questions as following:

- Page 10, Line 3 "Population data were collected from the Shanxi Statistical...." it is control group, right? if so, you have to give more details about sampling inclusion criteria/ sampling technique.

- Page 10, It is mentioned that the diagnosis done according to the "National Diagnostic Criteria for Visceral

Leishmaniasis in China." would you please add a reference for that.

**Results**

-Does the analysis presented match the analysis plan?

-Are the results clearly and completely presented?

-Are the figures (Tables, Images) of sufficient quality for clarity?

Reviewer #1: (No Response)

Reviewer #2: Yes

Reviewer #3: - In page 13, the map, what is the indications for the other colours not mentioned in the map legend. (Yellow and blue)

- Table 1, did you go with the questionnaire with population less than 5 years old, 15 years old? please confirm.

- Table 2, the majority of cases did not hear about the disease before, it means that they are not aware of their diagnosis or there is different between the name in the survey and the one use in the community? please explain. 

- Table 2, why did you include the cases and control who responded about "have you heard about the MT-ZVL" if they say "No" in the analysis of the rest of questions. then the answer after that might be randomly and not reflecting the reality, right? please explain

**Conclusions**

-Are the conclusions supported by the data presented?

-Are the limitations of analysis clearly described?

-Do the authors discuss how these data can be helpful to advance our understanding of the topic under study?

-Is public health relevance addressed?

Reviewer #1: (No Response)

Reviewer #2: Yes

Reviewer #3: - In the limitation section. we may discuss that the time frame of cases was very wide and that might affect the knowledge as well as the living condition including the risk factors. 

- Also, the mobility of cases over 15 years can do the same.

- Page 20, The author mentioned "The awareness of MT-ZVL knowledge was 59.8% ..." Would you explain how it is calculated as it was not mentioned before and not presented in the results.

**Editorial and Data Presentation Modifications?**

Reviewer #1: (No Response)

Reviewer #2: Dear Author,

Would you please confirm that you interviewed the patients and control at the same time?

Reviewer #3: (No Response)

**Summary and General Comments**

Reviewer #1: In this manuscript, the variables have been chosen correctly and their analysis has been done accurately. But it has no novelty, and in my opinion, it does not have the necessary quality to be published in Plus One journal.

Reviewer #2: In general it is fine

Reviewer #3: Thank you very much for the effort done in the paper and the level of scientific investigations for answering the objective of the study. Generally, the paper is well organized and homogenous even there was some comments to be answered mentioned above in different sections as well as some common points mentioned below

- In the Abstract methods. ot is confusing for me particularly with control part

- In the Abstract results. it would be better to quantify the results mentioned (Adding numbers) to the narrative papagrapg.

- In introduction, in the second paragraph, please add (In China) as you were talking globally just in the previous sentences. 

- Regarding the declaration of ethics approval and consent to participate. this research has 2 parts of data collection. The first part is 2ry analysis and the second part is active data collection. Please double check if there is no need for IRB approval.
---

## [Decision Letter · Decision Letter 1]

30 Apr 2024

Dear Dr. Li,

We are pleased to inform you that your manuscript 'Re-emergence and influencing factors of mountain-type zoonotic visceral leishmaniasis in the extension region of Loess Plateau, China' has been provisionally accepted for publication in PLOS Neglected Tropical Diseases.

Best regards,

Johan Van Weyenbergh

Academic Editor

Reviewer's Responses to Questions

**Key Review Criteria Required for Acceptance?**

**Methods**

-Are the objectives of the study clearly articulated with a clear testable hypothesis stated?

-Is the study design appropriate to address the stated objectives?

-Is the population clearly described and appropriate for the hypothesis being tested?

-Is the sample size sufficient to ensure adequate power to address the hypothesis being tested?

-Were correct statistical analysis used to support conclusions?

-Are there concerns about ethical or regulatory requirements being met?

Reviewer #2: Yes

**Results**

-Does the analysis presented match the analysis plan?

-Are the results clearly and completely presented?

-Are the figures (Tables, Images) of sufficient quality for clarity?

Reviewer #2: yes

**Conclusions**

-Are the conclusions supported by the data presented?

-Are the limitations of analysis clearly described?

-Do the authors discuss how these data can be helpful to advance our understanding of the topic under study?

-Is public health relevance addressed?

Reviewer #2: yes

**Editorial and Data Presentation Modifications?**

Reviewer #2: (No Response)

**Summary and General Comments**

Reviewer #2: (No Response)

PLOS authors have the option to publish the peer review history of their article (what does this mean?). If published, this will include your full peer review and any attached files.

Reviewer #2: No

---

## [Editor Report · Acceptance letter]

26 May 2024

Dear Dr. Li,

We are delighted to inform you that your manuscript, "Re-emergence and influencing factors of mountain-type zoonotic visceral leishmaniasis in the extension region of Loess Plateau, China," has been formally accepted for publication in PLOS Neglected Tropical Diseases.

Best regards,

Shaden Kamhawi

co-Editor-in-Chief

Paul Brindley

co-Editor-in-Chief
